# Retinal Ischemia: Therapeutic Effects and Mechanisms of Paeoniflorin

**DOI:** 10.3390/ijms262210924

**Published:** 2025-11-11

**Authors:** Windsor Wen-Jin Chao, Howard Wen-Haur Chao, Pai-Huei Peng, Yi-Tzu Lee, Hsiao-Ming Chao

**Affiliations:** 1Department of Renal Medicine, New Cross Hospital, Wolverhampton WV10 0QP, UK; windsor.chao123@gmail.com; 2Department of Science, University of British Columbia, Vancouver, BC V6T 1Z4, Canada; 3Department of Renal Medicine, Queen Elizabeth University Hospital, Birmingham B15 2TH, UK; wenhaur.chao@gmail.com; 4Department of Ophthalmology, Shin Kong Hospital, Taipei 11101, Taiwan; 5Institute of Pharmacology, Department of Medicine, School of Medicine, National Yang Ming Chiao Tung University, Taipei 11221, Taiwan; s851009@yahoo.com.tw; 6Department of Emergency, Taipei Veterans General Hospital, Taipei 11217, Taiwan; 7Department of Chinese Medicine, School of Chinese Medicine, China Medical University, Taichung 40402, Taiwan

**Keywords:** retinal ischemia, paeoniflorin, oxidative stress, beta-catenin, angiopoietin-2, pigment epithelium-derived factor, antioxidation, traditional Chinese medicine, VEGF

## Abstract

Retinal ischemia is a key factor in the progression of vision-threatening ocular diseases, including central retinal artery/vein occlusion, exudative age-related macular degeneration (eAMD), and proliferative diabetic retinopathy. This study investigates the effects of paeoniflorin along with its related neuroprotective molecular pathways in the treatment of retinal ischemia. Free radical or ischemic-like damage was induced by incubating retinal pigment epithelium (RPE) cells for 24 h with 1 mM hydrogen peroxide (H_2_O_2_) or by subjecting retinal neuronal cells to 8 h of oxygen–glucose deprivation (OGD). Both treatments caused significant cell loss. Treatment with paeoniflorin significantly increased cell viability at 0.5 mM in both cell types. In a Wistar rat model of retinal ischemia and reperfusion (I/R) elicited by sustained high intraocular pressure (HIOP), pre-treatment with 0.5 mM paeoniflorin mitigated the ischemia-induced decline in ERG b-wave amplitude, reduction in whole and inner retinal thickness, loss of fluorogold-labeled retinal ganglion cells, and formation of apoptotic cells. Meanwhile, paeoniflorin effectively downregulated pro-neovascular mediators β-catenin, hypoxia-inducible factor 1-alpha (HIF-1α), vascular endothelial growth factor (VEGF), and the pro-inflammatory/angiogenic biomarker angiopoietin-2 (Ang-2), producing effects similar to the Wnt/β-catenin inhibitor (dickkopf-related protein 1), anti-angiogenic pigment epithelium-derived factor (PEDF), and anti-VEGF Avastin (bevacizumab). These findings suggest that paeoniflorin may protect against retinal ischemia through its anti-inflammatory, anti-neovascular/angiogenic, antioxidative, and neuroprotective properties.

## 1. Introduction

Retinal ischemia is a leading cause of vision loss worldwide impacting millions of people [1]. It serves as a common underlying pathology in sight-threatening conditions such as branch or central retinal vein occlusion, normal-tension glaucoma, and exudative age-related macular degeneration (eAMD) [1,2]. Epidemiological studies report that the prevalence of branch retinal vein occlusion, central retinal vein occlusion, and any retinal vein occlusion among individuals aged 30 years and older is estimated at 0.64% (23.38 million people), 0.13% (4.67 million people), and 0.77% (28.06 million people), respectively [3]. Furthermore, AMD alone is expected to affect nearly 288 million people globally by 2040, accounting for 8.7% of blindness cases worldwide and representing the predominant cause of vision loss in economically developed regions [4]. These data highlight the substantial global burden of retinal ischemia and underscore the pressing need to develop targeted treatment strategies to reduce its impact across affected populations.

Retinal ischemia-related ocular disorders are strongly associated with the long-term accumulation of free radicals [5,6], driven by pathological processes at the cellular level. A notable example involves retinal ganglion cells (RGCs), which are highly specialized relay neurons with limited regenerative capacity, rendering them more susceptible to oxidative injury than other retinal neurons [7]. Once non-viable, these cells are generally not replenished [7]. Their death can activate surrounding microglia and other retinal cells, leading to the degeneration of adjacent neurons and severe disruption to the transmission of visual signals from the retina to the brain [8]. Another example are the retinal pigment epithelial (RPE) cells which, although epithelial in nature, are an essential component of the retina. They form a single layer of cuboidal cells beneath rod and cone photoreceptors and play a critical role in visual function by facilitating photoreceptor renewal through ingestion and lysosomal degradation of shed outer segments. RPE cells are post-mitotic with limited regenerative ability, and their death adversely affects photoreceptor survival [9]. Progressive loss of RPE cells contributes to retinal degenerative diseases such as AMD. Another important role of RPE cells in the context of retinal ischemia is the secretion of pigment epithelial-derived factor (PEDF) [10], a potent inhibitor of abnormal angiogenesis that prevents pathological neovascularization [11]. Consequently, the progressive loss of either RGC or RPE cells due to retinal ischemia and reperfusion (I/R) injury can result in significant visual acuity impairment, leading to the onset of retinal degenerative conditions, such as eAMD. Individuals affected by ischemia-related ocular disorders often suffer from profound visual deficits, including impaired dark adaptation, photopsia, and metamorphopsia [12,13]. Moreover, poor visual outcomes may lead to psychological distress and depression [9], underscoring the considerable impact on both functional vision and overall quality of life.

It is well established that the Wnt/β-catenin pathway serves as an important upstream regulator in the signaling cascade that drives vascular endothelial growth factor (VEGF) activation during ischemic injury [14,15,16,17]. Specifically, Wnt/β-catenin controls the expression of hypoxia-inducible factor 1-alpha (HIF-1α), which functions as an intermediate downstream mediator. Evidence from Lee et al. (2009) shows that inhibition of Wnt/β-catenin markedly reduces HIF-1α levels, confirming that HIF-1α expression depends on upstream Wnt/β-catenin activity [18]. In turn, HIF-1α directly stimulates VEGF transcription, making VEGF the final downstream effector responsible for pathological angiogenesis. This sequential activation—from Wnt/β-catenin to HIF-1α and finally to VEGF—results in the growth of weak, deformed neovascular vessels prone to rupture. Beyond angiogenesis, ischemia triggers intense inflammatory responses and reperfusion injury which further amplifies this inflammation, ultimately causing apoptosis of retinal cells such as RGCs and RPEs [19]. Overall, the overexpression of β-catenin, HIF-1α, and VEGF biomarkers in retinal cells is strongly associated with the progression of retinal ischemia-related disorders, as discussed earlier.

In addition to the Wnt/β-catenin-associated VEGF pathway, angiopoietin-2 (Ang-2) is another vasoactive factor implicated in oxidative stress and plays a key role in regulating retinal vascular stabilization and remodeling [20]. Ang-2 is overexpressed in endothelial cell Weibel–Palade bodies during ischemic and inflammatory conditions, in contrast to angiopoietin-1 (Ang-1), a homeostatic marker that remains relatively constant under all conditions [20,21]. This is supported by human vitreous extraction studies showing elevated levels of Ang-2 in patients with eAMD, proliferative diabetic retinopathy, and retinal artery/vein occlusion, whereas Ang-1 remains at baseline concentrations in both healthy and diseased eyes [22]. In this case, the overexpression of Ang-2 is believed to disrupt Ang-1/Tie2 signaling, resulting in blood vessel destabilization (e.g., pericyte loss), increased inflammatory cytokine levels (e.g., MCP-1), and weakened endothelial cell junctions, contributing to vascular dysfunction observed in retinal and choroidal vascular diseases [20]. Moreover, Ang-2 is recognized as an inflammatory mediator that enhances the effects of VEGF [23]. In a study by Benest et al. (2013), Ang-2 deficient mice exhibited reduced vascular leakage in response to VEGF, while wild-type mice experienced increased leakage with VEGF treatment and no change when treated with saline [24]. These findings suggest that Ang-2 is essential for VEGF-induced vascular permeability. Furthermore, studies indicate that co-inhibiting Ang-2 and VEGF pathways significantly reduces blood vessel leakage and tumor-associated angiogenesis, offering a synergistic benefit over single-agent therapy with no substantial increase in systemic toxicity [25]. Preliminary human trials also show that dual-pathway inhibition is more effective at reducing neovascularization in patients with eAMD, diabetic macular edema, and retinal artery/vein occlusion, compared to anti-VEGF monotherapy [26].

*Paeonia lactiflora* Pallas is a well-known traditional herbal remedy used across China, Korea, and Japan for the treatment of pain, inflammatory, and auto-immune disorders, making it a promising natural product for further investigation into its therapeutic potential [27,28,29,30]. The purpose of this paper is to explore the therapeutic properties of paeoniflorin (C_23_H_28_O_11_), a water-soluble monoterpene glucoside extracted from the roots of *Paeonia lactiflora* Pallas using solvent or subcritical water extraction methods [31,32]. Paeoniflorin is the main bioactive component of this herb, which accounts for more than 90% of the total glucoside content, and is largely responsible for the therapeutic effects demonstrated in vitro and in vivo investigations [31,33], including free radical scavenging, anti-inflammatory, and neuroprotective properties [27,28,29]. For instance, paeoniflorin has been shown to downregulate the Wnt/β-catenin pathway involved in apoptosis and inflammation in coronary artery disease [34]. Similarly, Zhang and Wei (2020) reported that paeoniflorin inhibited the Wnt/β-catenin pathway via reducing Wnt-1 and β-catenin mRNA and protein levels, thereby alleviating diabetes-associated renal damage caused by oxidative stress [35]. In hepatocellular carcinoma cells, paeoniflorin downregulates 5-HT1D, attenuating β-catenin pathway components [36]. In colorectal cancer, paeoniflorin inhibits miR-3194-5p, relieving the repression of CTNNBiP1, a negative modulator of β-catenin, resulting in pathway deactivation [37]. Given these findings, it would be valuable to investigate whether paeoniflorin can downregulate the upstream biomarker Wnt/β-catenin in the context of retinal ischemia, along with associated downstream biomarkers such as HIF-1α and VEGF, as well as the pro-inflammatory/angiogenic marker Ang-2. Such an investigation would highlight its potential therapeutic relevance. These protective effects of paeoniflorin remain underexplored in retinal ischemia, underscoring the need for further research.

Using cell cultures exposed to hydrogen peroxide (H_2_O_2_) and oxygen–glucose deprivation-induced (OGD) to induce ischemic-like damage, as well as an in vivo model of I/R injury established by high intraocular pressure (HIOP), the antioxidative, anti-inflammatory, and anti-angiogenic properties of paeoniflorin were evaluated. Established treatments such as anti-VEGF therapy and steroids may be inadequate in visual outcomes in advanced cases of retinal ischemic-related ocular disorders, despite controlling hemorrhage and foveal swelling [38]. Consequently, for the management of ischemic disorders (e.g., eAMD), early use of antioxidants derived from natural products—either for prevention or as adjunctive therapy—is often considered, highlighting the importance of alternative and complementary medicines in chronic disease management [13,39]. Novel agents (e.g., paeoniflorin) which can downregulate upstream signaling pathways of VEGF (e.g., Wnt/β-catenin) or reduce Ang-2 levels are currently being explored as alternative therapeutic strategies for ischemic retinal disorders, particularly in cases where conventional treatments have proven ineffective.

In the present study, multiple research techniques were employed to assess the therapeutic potential of paeoniflorin, such as cell culture assays, electrophysiological studies, and mRNA analysis using real-time polymerase chain reaction (RT-PCR). Additional histopathological methods included retrograde fluorogold immunolabeling of RGCs, hematoxylin and eosin (HE) staining, and terminal deoxynucleotidyl transferase (TdT)-mediated dUTP nick end labeling (TUNEL) assay. It is hypothesized that paeoniflorin may protect against retinal ischemia through its neuroprotective and antioxidative, and anti-inflammatory effects via the downregulation of pro-angiogenic β-catenin, HIF-1α, VEGF, and inflammatory Ang-2 biomarkers with similar effectiveness to Wnt/β-catenin inhibitor dickkopf-related protein 1 (DKK1), anti-angiogenic PEDF, and anti-VEGF therapy.

## 2. Results

### 2.1. Free Radical-Induced Cellular Damage (In Human RPE Cells)

Cellular viability ratio was assessed by counting the number of ARPE-19 cells under optical microscopy (Figure 1; *n* = 4). Paeoniflorin 0.25 or 0.5 mM was added 15 min before exposure to 1 mM H_2_O_2_ for 24 h. As shown in Figure 1, the normal group (A/E)—the human RPE cells incubated in DMEM [median, 100.25% (Q25–Q75: 99.47–100.28%)]—exhibited the highest ratio of cell viability, in contrast to the H_2_O_2_-treated group [34.94% (21.14–40.00%)] (B/E). In this case, the ARPE-19 cells exposed to 1 mM H_2_O_2_-induced free radical damage had significantly (^†^
*p* < 0.05) lower cell viability. Conversely, Figure 1B–D indicates pre-treatment for 15 min with 0.25 mM [51.76% (31.54–78.95%)] and 0.5 mM [96.70 (87.74–103.60)]. Paeoniflorin produced a significant therapeutic effect (^†^
*p* < 0.05) at 0.5 mM of paeoniflorin against free radical injury, resulting in higher ARPE-19 cell viability, comparable to the control group in DMEM. However, 0.25 mM paeoniflorin did not confer any significant protective effects.

### 2.2. Oxygen Glucose Deprivation (In RGCs)

RGC count was assessed using light microscopy (*n* = 6). For the RGC study, 0.25 and 0.5 mM paeoniflorin were administered 15 min prior to the OGD insult. As shown in Figure 2A,E, compared with the control groups [median, 100.76% (Q25–Q75: 97.46–102.74%)], OGD-induced oxidative stress resulted in significant (^†^* p* < 0.05) reduction in cell viability [58.82% (52.86–60.80%)]. Similar to the ARPE-19 results, pre-treatment with 0.25 mM [62.01% (56.83–64.34%); Figure 2C] and 0.5 mM [75.70% (73.40–80.90%); Figure 2D] paeoniflorin produced no significant effect at 0.25 mM but showed a significant neuroprotective effect at 0.5 mM (^†^
*p* < 0.05). In other words, higher cell viability was observed at 0.5 mM of paeoniflorin, in terms of lowering ischemic-like insult.

### 2.3. Electroretinogram: The Effect of Pre-Ischemic Treatment Paeoniflorin on Ischemic–Reperfusion Injury

The ERG wavefront averages were utilized to examine the retinal electrophysiological response (*n* = 8). As shown in Figure 3A, there was a substantial reduction in ERG b-wave amplitude following I/R injury in the vehicle-pre-treated group, labeled as the vehicle + I/R group. Specifically, there was a significant (^†^
*p* < 0.05) reduction in the b-wave ratio in the vehicle + I/R group [median 7.13% (Q25–Q75: 2.46–11.05%)] at 24 h following ischemia, relative to the normal group [104.45% (84.08–113.32%)]. Importantly, pre-ischemic administration of paeoniflorin at both 0.25 mM and 0.5 mM attenuated this reduction. Notably, a significant (^†^
*p* < 0.05) protective effect was observed at 0.5 mM paeoniflorin, which mitigated the b-wave amplitude decrease. In contrast, 0.25 mM paeoniflorin did not exert a significant therapeutic effect.

### 2.4. Fluorogold Labeling and Density of RGCs

The RGC density was evaluated in the following groups: the normal group (Figure 4A), the vehicle-pre-treated ischemic group (Figure 4B), and the groups pre-treated with 0.25 mM (Figure 4C) or 0.5 mM paeoniflorin group (Figure 4D). As shown in Figure 4E, there was a significant (^†^ *p* < 0.05) difference between the normal group [median 97.72% (Q25–Q75: 82.20–120.06%)] and the vehicle + I/R group [49.04% (41.50–61.57%)], indicating increased RGC mortality after I/R. Further analysis also showed a significant difference (^†^ *p* < 0.05) between the vehicle + I/R and the 0.5 mM paeoniflorin + I/R group [168.41% (78.74–185.47%)], confirming the protective effect of paeoniflorin. However, 0.25 mM paeoniflorin + I/R group [55.97% (46.17–65.02%)] did not exhibit a significant inhibitory effect against I/R.

On the other hand, morphometric analysis of the retinal thickness is presented in Figure 4F,G. Compared with the normal group (Group 1), the thicknesses of both the whole retinal layer and inner retinal layer were significantly reduced in the vehicle-treated ischemic retinas (Group 2). The reductions were attenuated in a dose-dependent manner in the groups pre-treated with 0.25 mM (Group 3) and 0.5 mM of paeoniflorin (Group 4). Results are expressed as the median value with 25% and 75% quartiles (*n* = 8) and significant differences are represented as (^†^ *p* < 0.05).

Quantitative analysis of apoptotic cells (TUNEL-positive) is shown in Figure 4H. A significant increase (^†^
*p* < 0.05) in apoptotic cells was observed in the vehicle + I/R group compared with the normal group. Pre-ischemic treatment with 0.5 mM paeoniflorin significantly (^†^
*p* < 0.05) reduced the number of apoptotic cells. Results are expressed as the median value with 25% and 75% quartiles (*n* = 6).

### 2.5. Measurement of β-Catenin, HIF-1α, VEGF, and Ang-2 in Primary Retinal Cells via Real Time Polymerase Chain Reaction (RT-PCR)

In Figure 5A–D (*n* = 6), mRNA levels of β-catenin, HIF-1α, VEGF, and Ang-2 were analyzed in retinal cells exposed to I/R, with or without paeoniflorin pre-treatment, and compared to the Wnt/β-catenin inhibitor DKK1, anti-angiogenic PEDF, and anti-VEGF agent Avastin. Various mRNA biomarkers from primary retinal cells (normal control) were analyzed and demonstrated expected baseline expression levels, as presented in Figure 5A [β-catenin; median 93.94 (Q25–Q75: 90.57–115.05)], Figure 5B [HIF-1α; 98.90 (97.86–102.96)], Figure 5C [VEGF; 89.32 (72.16–130.81)], and Figure 5D [Ang-2; 99.68 (91.55–106.36)].

When comparing the normal group with retinal cells subjected to I/R and pre-treated with vehicle (vehicle + I/R), the vehicle + I/R group showed a significant (^†^
*p* < 0.05) upregulation of β-catenin [247.44 (224.05–296.64)], HIF-1α [787.35 (738.84–874.41)], VEGF [477.94 (287.80–631.28)], and Ang-2 [640.82 (586.89–778.81)]. This elevation was not significantly ameliorated by 15 min pre-treatment with 0.25 mM of paeoniflorin. However, the most significant reduction (^†^
*p* < 0.05) was seen with pre-administration of 0.5 mM paeoniflorin, in regard to the biomarkers β-catenin [95.56 (68.12–110.20)], HIF-1α [61.38 (52.72–72.55)], VEGF [76.19 (47.44–125.20)], and Ang-2 [95.40 (89.70–103.64)].

Importantly, 0.5 mM paeoniflorin showed similar and significant (^†^
*p* < 0.05) therapeutic effectiveness to 1 μg/4 μL of the Wnt/β-catenin inhibitor DKK1 and 100 μg/4 μL anti-VEGF Avastin, as observed in Figure 5A [DKK1 + I/R; 74.05 (53.56–95.27)] and Figure 5C [Avastin 100 μg/4 μL; 32.93 (22.62–72.81)], in downregulating β-catenin and VEGF expression. In the context of the HIF-1α/β-actin ratio (Figure 5C), 0.5 mM paeoniflorin demonstrated even greater effectiveness than 100 μg/4 μL of Avastin in reducing HIF-1α, an intermediate biomarker involved in VEGF expression. These results collectively highlight the multifaceted mechanisms of action of paeoniflorin in counteracting I/R injury, providing advantages over single-pathway drugs (e.g., Avastin).

### 2.6. Comparison of PEDF (Anti-Angiogenic Agent) Versus Paeoniflorin on VEGF Expression Using mRNA Analysis

In Figure 5 (*n* = 6), VEGF mRNA expression in retinal cells exposed to I/R (HIOP model) pre-treated with vehicle (vehicle + I/R) was compared to normal retinal cells [median 102.05 (Q25–Q75: 88.40–109.62)]. The vehicle + I/R group [623.22 (489.00–836.61)] exhibited significant (^†^
*p* < 0.05) increase in pro-angiogenic VEGF mRNA levels. This elevation was significantly (^†^
*p* < 0.05) attenuated by 15 min of pre-treatment with 0.5 mM paeoniflorin [104.44 (99.76–107.16)], showing similar significant therapeutic effectiveness to anti-angiogenic agent PEDF [104.85 (89.35–113.09)]. Both 0.5 mM paeoniflorin and PEDF achieved approximately six-fold reduction in VEGF levels relative to the vehicle + I/R group.

## 3. Discussion

Ischemia is a critical driving factor in the progression of several retinal diseases, including eAMD, CRAO, diabetic retinopathy, and other related ocular disorders. Over the past two decades, treatments such as radiation, transpupillary thermotherapy, photodynamic therapy, and anti-VEGF agents have proven clinically useful in managing these conditions [9,15]. While hemorrhage and fluid buildup have been effectively controlled, some individuals still experience suboptimal visual outcomes despite these treatments [35]. Although clinical trials of intravitreal anti-VEGF injections have demonstrated significant improvements in visual outcomes, clinical data suggest that these benefits may not always translate into sustained or optimal results for all patients, likely due to the complexity of disease pathophysiology and the limitations of current therapies [15]. Consequently, ongoing research is focusing on exploring new approaches that target multiple pathogenic pathways. These include upstream inhibitors of the Wnt/β-catenin pathway, which block the overexpression of downstream neovascular biomarkers such as HIF-1α and VEGF, as well as drugs that suppress the inflammatory amplifier/modulator Ang-2 [33]. There is an increasing demand for therapies that address the complex and multifactorial causes of retinal vascular diseases beyond VEGF, with the goal of improving outcomes, providing sustained efficacy, and reducing the need for frequent treatments.

Presently, an investigation has been conducted to determine whether paeoniflorin exhibits neuroprotective effects against H_2_O_2_, OGD, and HIOP-induced I/R injury in retinal cells, such as RPEs and RGCs. This is particularly relevant given the well-established role of oxidative stress in retinal ischemia. Notably, the current findings regarding paeoniflorin’s neuroprotective effects could contribute to novel strategies for the prevention and treatment of ischemic eye disorders. Paeoniflorin has been identified in numerous studies as an effective free radical scavenger and cytoprotective agent [36], acting by inhibiting the formation of intracellular reactive oxygen species (ROSs) and apoptotic caspase-3 activation in RPE cells, while downregulating the phosphorylation of ERK and p38 MAPK [37]. A preliminary study demonstrated that paeoniflorin protected against I/R injury by preserving ERG a- and b-wave amplitudes (supported by Figure 3) but did not identify the underlying mechanism [38]. Therefore, it is essential to explore the protective activity of paeoniflorin in relation to upstream Wnt/β-catenin and downstream HIF-1α/VEGF/Ang-2 pathways, which are known to play vital roles in ischemia-triggered angiogenic processes, including leukocyte adhesion, endothelial cell survival, vascular stabilization, and retinal neovascularization [12]. The results obtained from these studies could offer valuable alternative and complementary therapies to prevent the progression and worsening of retinal ischemic disorders (e.g., eAMD), which are associated with significant complications and comorbidities. Consequently, early and timely intervention in retinal ischemia is crucial for optimizing visual outcomes and preserving vision in affected individuals.

Research has shown that the rat retina, with its well-perfused vascular system, is analogous to the human retina, making it a preferred model for studying retinal ischemia over animals like cats, rabbits, or monkeys [26]. Thus, rats were chosen in this study to investigate retinal ischemia due to their ability to provide reproducible results that are relevant to human disease as well as their practicality [26]. As for our extracted retinal cells, immunolabelling and RT-PCR confirmed Thy-1 expression, reinforcing the validity of the cells utilized. As shown in our previous publications [34,40], Thy-1 is localized to the inner plexiform layer (IPL) and ganglion cell layer (GCL) of rat RGCs, confirming the expression of typical RGC markers as described in the literature and validating the authenticity of our extracted RGCs. By combining in vitro and in vivo methods, we enhanced the reliability and translational value of the study. In regard to the limitations of the RPE cells, it should be acknowledged that retina is composed of a complex multicellular network, hence the RPE cells alone do not sufficiently represent overall retinal cell viability. Also, the culture of RPE cells and exposure to H_2_O_2_ would not likely represent a pathological event in vivo, especially after isolating RPE cells only. This is because the neuroretina is generally more sensitive than RPE cells, and isolating and exposing them to H_2_O_2_ may not fully mimic pathological findings in the neuroretina. Hence, experiments using oxygen glucose deprivation/reoxygenation (OGD/R) in neuronal cells (RGCs) to simulate cell I/R injury models have been included, as oxygen glucose deprivation replicates the energy deficit experienced in ischemia, followed by reoxygenation to mimic reperfusion. Also, this new data provides direct neuronal evidence linking in vitro findings to in vivo outcomes. However, it is important to note that no model can fully replicate the complexity of retinal ischemia, a multifaceted neurodegenerative disorder. Despite ongoing challenges in fully understanding diseases like AMD, these models continue to provide valuable insights into retinal ischemia.

As highlighted in prior research and our previous discussion, ROS and oxidative stress play a significant role in the onset of diseases such as eAMD, including the upregulation of inflammatory enhancers and modulators such as MCP-1 [41,42], Ang-2 [12], IL-13 [6], and others. In the present study, the in vivo animal model showed that 1 mM H_2_O_2_ and OGD induced significant free radical injury, resulting in cellular mortality (Figure 1B,E and Figure 2B,E), compared with the control group of cells grown in DMEM (Figure 1A,E and Figure 2A,E). As shown in Figure 1 and Figure 2, 15 min pre-treatment with paeoniflorin successfully reduced cellular mortality, showing a significant effect (^†^
*p* < 0.05; Figure 1E and Figure 2E) at 0.5 mM concentration, whereas no significant effect was observed at the lower dosage of 0.25 mM (Figure 1C and Figure 2C). In essence, this study demonstrates that paeoniflorin can protect against ROS-induced toxicity (Figure 1D,E and Figure 2D,E). These data further suggest that paeoniflorin may act as a free radical scavenger, potentially inhibiting the progression of oxidative stress-related retinal pathologies and offering therapeutic value in conditions such as eAMD.

From a therapeutic standpoint, the protective effect of paeoniflorin may be attributed to the downregulation of upstream β-catenin along with downstream HIF-1α, VEGF, and Ang-2 levels (Figure 5A–D). By suppressing these ischemia-related biomarkers, paeoniflorin appears to counteract HIOP-induced I/R injury, confirming its anti-ischemic properties. RT-PCR results further validate that 0.5 mM paeoniflorin protects against I/R by exerting a similar effect to that of PEDF, a naturally produced cytokine involved in mitigating pathological angiogenesis, via antagonizing VEGF effects induced in I/R [5,6]. In patients with BRVO and CRVO, PEDF levels are reported to be lower than in healthy individuals and are inversely correlated with retinal thickness [43]. As shown in Figure 5E, the addition of either 0.5 mM paeoniflorin or 2 μg PEDF resulted in approximately a six-fold decrease in VEGF levels with no significant difference between the two treatments, relative to the vehicle + I/R group. Similarly, mRNA analysis shows that the commercially purchased 100 μg/4 μL anti-VEGF monoclonal antibody (Avastin) exhibited the same effectiveness as 0.5 mM paeoniflorin in downregulating VEGF levels (Figure 5C). However, no significant reduction in HIF-1α mRNA expression was observed with Avastin (Figure 5B). Since Avastin is a monoclonal antibody designed to neutralize VEGF directly, the absence of upstream HIF-1α inhibition is expected and consistent with its known mechanism of action [15]. Finally, a similar pattern can also be observed when comparing the control experimental group DKK1 (a negative regulator of the Wnt/β-catenin signaling pathway) with 0.5 mM paeoniflorin, in terms of downregulation of upstream β-catenin biomarker levels in I/R (Figure 5A). Collectively, these findings demonstrate the multi-mechanistic properties of paeoniflorin in combating retinal ischemia, in contrast to single-pathway therapies such as anti-VEGF.

Importantly, mRNA analysis demonstrates that pre-treatment with 0.5 mM paeoniflorin significantly downregulates Ang-2 (Figure 5D), a pro-inflammatory cytokine expressed under ischemic conditions [12], supporting paeoniflorin’s additional anti-inflammatory role. This effect can be attributed to Ang-2’s ability to modulate and amplify inflammation by promoting macrophage/microglia accumulation [15] and by sensitizing endothelial cells to TNF-α, bradykinin, and histamine [12]. Furthermore, Ang-2 enhances the response of retinal vessels to VEGF-A [16], thereby increasing vascular permeability and neovascularization, as shown in double transgenic mice with inducible VEGF-A expression in the retina [44]. In a separate study, combined Ang-2/VEGF-A inhibition led to a significant reduction in retinal inflammation in JR5558 mice, evidenced by a marked decrease in Iba1-plus microglia/macrophages surrounding CNV lesions, compared to the effects of inhibiting Ang-2 or VEGF-A alone [15]. In our study, we also observed this dual-pathway mechanism, where paeoniflorin effectively targets both VEGF (Figure 5C) and Ang-2 (Figure 5D) pathways, highlighting its potential for synergistic therapeutic effects in retinal ischemia-related disorders. Thus, reducing the levels of these markers may help prevent the exacerbation of vessel leakage, inflammation, and neovascularization, processes that are heavily involved in retinal ischemia. Both prior and current studies consistently identify elevated levels of Wnt/β-catenin, HIF-1α, VEGF, and p38 mitogen-activated protein kinase [35,45], as well as Ang-2 (Figure 5) in I/R injury and various retinal ischemia-driven disorders (e.g., eAMD). Based on these findings, paeoniflorin may offer an approach that complements and enhances conventional treatments for mitigating oxidative stress and visual deterioration induced by retinal ischemia and may reduce treatment frequency compared to monotherapies.

Earlier research has demonstrated that post-treatment with paeoniflorin counteracts the loss of ERG b-wave ratios caused by HIOP damage [38]. In this context, HIOP-induced I/R leads to impaired neuronal physiological function, as evidenced by a significant reduction in ERG amplitude (e.g., b-wave) when comparing the I/R group to the control group, which is consistent with the findings in Figure 3. In the present study, ERG results confirm that pre-treatment with 0.5 mM paeoniflorin significantly mitigates the ischemia-induced decline in the ERG b-wave. Beyond its ability to protect retinal electrophysiological function, pathological fluorogold labeling of RGCs in the in vivo studies revealed significantly reduced cell mortality following ischemia after pre-administration of 0.5 mM paeoniflorin (Figure 4). As previously mentioned, RGCs play a critical role in transmitting light stimuli from the retina to the occipital cortex; thus, their degeneration or loss can greatly disrupt the visual pathway and potentially lead to persistent visual impairment [46]. Both numerical and morphological analyses demonstrated that pre-treatment with 0.5 mM paeoniflorin effectively mitigated ischemia-induced cell death (Figure 4D,E), showing a significant contrast with the vehicle + I/R group (Figure 4B,E) and a pattern similar to the normal group (Figure 4A,E).

In this case, the HIOP model was employed to induce I/R injury and to obtain data on the therapeutic effects of paeoniflorin against retinal ischemia. This is a popular ischemic model for understanding the pathophysiology involved in transient retinal ischemia and the subsequent recovery mechanisms. Hence, care should be exercised when extrapolating these findings to chronic conditions [25]. In other words, one should recognize that this model only emulates acute I/R, with limitations in fully translating findings to humans [26]. The previously noted considerations can be seen as potential drawbacks and limitations of this study. Other extra-surgical factors prior to surgery can affect the studied compound’s potency and even mortality, such as continuous corneal lubrication, proper anesthesia, and controlled warming of the anesthetized rat, and so on [25]. Proper attention to these factors, as was done in this study, facilitates the creation of a reliable and standardized I/R injury model [25]. Though there are limitations, the HIOP model remains anatomically accurate and continues to be an essential tool for studying retinal ischemia. It offers a dependable approach for inducing, reversing, and quantifying ischemic damage solely to the retina, unlike other retinal ischemic models (e.g., ligation) that cause additional non-retinal damage to the ocular adnexa. Hence, the HIOP model provides a more focused retinal injury, making it well-suited for research on vision-threatening retinal ischemic disorders [25,26]. Additionally, increasing evidence supports the use of the HIOP model to evaluate non-neuronal endpoints, including retinal inflammation, vessel damage, hemorrhage, and fluid accumulation [47,48].

Through cell culture, electroretinography, histopathology, and RT-PCR techniques, the detrimental effects of free radical and I/R injury on the retina were observed, with paeoniflorin demonstrating a protective role in reducing these ischemic changes. These ischemia-induced alterations include increased retinal cell death (i.e., ARPE-19), a reduction in ERG b-wave amplitude, decreased fluorogold-labeled RGC density, and upregulation of neovascular/angiogenic/inflammatory, including β-catenin, HIF-1α, VEGF, and Ang-2. The results suggest that 0.5 mM paeoniflorin administered before ischemic injury, may act as a potential adjunctive or alternative therapy for retinal ischemia, utilizing novel mechanisms such as β-catenin, HIF-1α, VEGF, and Ang-2 RNA downregulation, especially in cases where single-pathway therapies (e.g., anti-VEGF) prove ineffective.

## 4. Materials and Methods

### 4.1. In Vitro Studies

#### 4.1.1. Cell Culture

The hRPE (ARPE-19, ATCC CRL-2302) and RGC-5 (ATCC PTA-6600) cell lines were obtained from the American Type Culture Collection (ATCC, Manassas, VA, USA). The cells were cultured in flasks with Dulbecco’s Modified Eagle’s Medium (DMEM) and Ham’s F-12 medium (DMEM/F-12; a 1:1 mixture of DMEM and Ham’s F-12 nutrient mixture) supplemented with 1% penicillin (5000 U/mL), streptomycin (5 mg/mL), neomycin (10 mg/mL) solution (Sigma-Aldrich, St. Louis, MO, USA), and 10% fetal bovine serum (Gibco-Invitrogen Corporation, Carlsbad, CA, USA). The cells were cultured in a humidified atmosphere of 95% air and 5% CO_2_ at 37 °C for 1 day. Generally, 80% confluent cultures of cells from 75 cm^2^ filter-capped cell culture flasks were passaged at a ratio of 1:3. Trypsin–ethylenediaminetetraacetic acid (Sigma-Aldrich) was used to detach the cultured adherent cells from the flask. After centrifugation, a new culture medium was added. Cells were cultured in six-well plates with 3 × 10^5^ cells seeded per well [9]. Cell numbers and cellular morphology were then subsequently verified under an optical microscope.

#### 4.1.2. Free Radical-Induced Cellular Damage

The hRPE cells were cultured in DMEM at 37 °C for one day with or without the inclusion of H_2_O_2_, plus a 15 min pre-treatment with vehicle or paeoniflorin. The groups were categorized into the normal control group (DMEM only), the experimental control group (H_2_O_2_ 1 mM to simulate oxidative stress), and the treatment group (paeoniflorin 0.25 mM or 0.5 mM with H_2_O_2_ 1 mM). To evaluate the impact of paeoniflorin on oxidative stress, cell viability was compared among the various groups via a hemocytometer. Each experiment was carried out in triplicate.

#### 4.1.3. Oxygen–Glucose Deprivation

According to Wu et al. (2024), RGCs were subjected to oxygen–glucose deprivation (OGD) for a duration of 8 h within a controlled environment simulating ischemia, which entailed culturing the cells in Dulbecco’s Modified Eagle Medium (DMEM) without glucose, maintained at 37 °C and under 1% O_2_, 94% N_2_, and 5% CO_2_ (monitored via a Penguin incubator; Astec Company, Fukuoka, Japan) [49,50]. The treated cells were then divided into various groups, namely DMEM only (normal control), 60 min of pre-OGD application with vehicle (vehicle + OGD), 60 min of pre-OGD application with 0.25 mM paeoniflorin (paeoniflorin 0.25 mM + OGD), and 60 min of pre-OGD application with 0.5 mM paeoniflorin (paeoniflorin 0.5 mM + OGD). At the end of the 8 h OGD period, the cultured cells were transferred to fresh DMEM for another day. Cell viability assessment was conducted using MTT assays.

#### 4.1.4. 3-(4,5-Dimethylthiazol-2-yl)-2,5-diphenyltetrazolium Bromide Assay

MTT (0.5 mg/mL; Sigma-Aldrich, St. Louis, MO, USA) was then introduced to the 24-well plates containing the original 400 μL of medium containing 4 × 10^4^ cells and left to incubate for 3 h at 37 °C. The reduced MTT (blue formazan) was then dissolved in 200 μL dimethyl sulfoxide. After agitation, the optical density (OD) of the solubilized formazan was measured using an ELISA reader (Synergy H1 Multi-Mode Reader, BioTek Instruments, Santa Clara, CA, USA) at 562 nm. Cell viability was determined by comparing the OD values to those of the control (100%; cells cultured in DMEM).

### 4.2. In Vivo Studies

#### 4.2.1. Animals

The Institutional Animal Care and Use Committee at Cheng Hsin General Hospital (CHGH; Taipei, Taiwan; Approval No: CHIACUC 106-10) approved all the animal experiments, which were conducted in accordance with the Association for Research in Vision and Ophthalmology (ARVO) Statement for the Use of Animals in Ophthalmology and Vision Research. Ninety-six Wistar rats (National Laboratory Animal Center, Taipei, Taiwan), six weeks old, weighing 200–250 g, and with equal sex distribution were housed at 40–60% humidity and at 19–23 °C. These rats were maintained under a 12 h light/dark cycle with 12–15 air exchanges per hour. They had access to food and water ad libitum [23]. The animals were then randomly allocated equally into each of their respective groups, as described below.

#### 4.2.2. Chemicals and Drug Administration

For electrophysiological (*n* = 8), immunohistochemical (*n* = 10) and mRNA studies (*n* = 6), either drug treatments or vehicle controls were administered via an intravitreal route in different groups. Specifically, the animals were randomly assigned to the following groups: normal, pre-ischemic administration of vehicle (vehicle + I/R), and pre-ischemic administration of paeoniflorin groups (paeoniflorin 0.25 or 0.5 mM + I/R). For the biomarker analyses (i.e., RT-PCR), in some instances, pre-administration of anti-angiogenic factor PEDF (Sigma-Aldrich; 607410-75-3; St. Louis, MO, USA), anti-VEGF antibody Avastin (Bevacizumab 50 or 100 μg/4 μL; Roche, Basel, Switzerland), and Wnt3a inhibitor Dkk1 (1 μg/4 μL; Abcam Inc., Cambridge, UK; ab281791; 95% purity HPLC) was utilized as a comparison to the normal, vehicle, and 0.25 or 0.5 mM paeoniflorin (Sigma-Aldrich; 23180-57-6; St. Louis, MO, USA) groups. Of note, the rats in the vehicle + I/R group that were exposed to ischemia were pre-administered with an equal volume of vehicle as those in the treatment groups.

#### 4.2.3. Anesthesia and Euthanasia

Animal anesthesia was induced through an intramuscular injection of 100 mg/kg ketamine (Pfizer, New York, NY, USA) combined with 5 mg/kg xylazine (Sigma-Aldrich, St. Louis, MO, USA) to minimize animal suffering. Euthanasia, on the other hand, was performed by an intraperitoneal injection of sodium pentobarbital (>140 mg/kg; Health-Tech Pharmaceutical Co., Ltd., Taipei, Taiwan), in accordance with the Scientific Procedures Act (1986).

#### 4.2.4. Ischemia Induction

Following the administration of anesthesia, the Wistar rats were positioned within a stereotaxic frame. The anterior chamber of the eye under study was cannulated with a 30-gauge needle (Becton, Dickinson and Company, Franklin Lakes, NJ, USA) connected to an elevated balanced salt solution reservoir (BSS*^®^* PLUS; Alcon, Geneva, Switzerland). The goal was to increase the intraocular pressure (IOP) to 120 mmHg in 60 min followed by reperfusion to mimic retinal ischemia. The saline was placed 1.63 m superior to the eyes till the top of the saline reservoir, in order to create the 120 mmHg pressure [24]. The presence of a pale fundus with a whitening iris confirmed the induction of ischemia [25,26]. Following ischemia, the rats were rested on warming pads maintained at 37 °C. I/R was defined as ischemia followed by reperfusion for a duration of 24 h. No procedures were performed on the normal (control) eyes. Finally, the pre-ischemic administration of paeoniflorin against ischemia and its associated biomarker pathways was determined through electroretinography, fluorogold retrograde labeling, and RT-PCR, as described below.

#### 4.2.5. Electroretinogram Recording

ERG was performed one day after I/R and pre-treatment with paeoniflorin or vehicle. Before the ERG recordings, the Wistar rats underwent dark adaptation for over 8 h [23]. Pupils were dilated with topical 1% tropicamide and 2.5% phenylephrine (Alcon, Geneva, Switzerland), and topical 0.5% proparacaine (Alcon, Geneva, Switzerland) was applied to anesthetize the ocular surfaces. Immediately before ERG recording, anesthesia was administered. ERG measurements were obtained from all animals both before and after ischemic insult (with intravitreal injections of vehicle or paeoniflorin). A strobe positioned 2 cm distal to the rat’s eye provided a stimulus at a frequency of 0.5 Hz. Fifteen consecutive responses were collected at two second intervals and at 10 kHz, and the responses were amplified using a P511/regulated power supply 107/stimulator PS22 (Grass-Telefactor; Astro Nova, Brossard, QC, Canada). To facilitate the comparison, the b-wave ratio of the treated ischemic eye was compared and standardized to that of the untreated contralateral normal eye. As for the normal (control) eyes, no procedures were carried out. In terms of exclusion criteria, data points were excluded if the b-wave ratio exceeded 125% or fell below 75% of baseline values [23].

#### 4.2.6. Retrograde Labeling of Retinal Ganglion Cell

Fluorogold retrograde labeling was used to label RGCs [25]. Following anesthesia, the rats were secured in a stereotactic frame, and two small holes were made in the skull, positioned 1.5 mm lateral to the midline and 6 mm posterior to the bregma. Fluorogold (5%, 2 μL; Sigma-Aldrich) was injected through the burr holes via a micropipette at depths of 3.8 mm, 4.0 mm, and 4.2 mm below the skull [27,28,29]. Retinal ischemia induction was carried out on the studied eye 72 h following retrograde labeling. The animals were sacrificed one day after retinal I/R injury.

The eyes were subsequently extracted utilizing a suture marker at the 12 o’clock position to ensure proper anatomical orientation. Following dissection of the anterior segment and removal of the vitreous body, the whole retina was meticulously extracted and fixed with 4% paraformaldehyde for 60 min. Following fixation, the whole retina was mounted on a slide and separated evenly into four quadrants. The slides were allowed to air dry before being treated with DPx agent (Fluka Chemical, Ronkonkoma, NY, USA). Each quadrant of the retina was subdivided into three regions: central, middle, and peripheral, located at distances of 1, 2, and 3 mm from the optic disk, respectively. Within each region, six microscopic areas of 0.430 × 0.285 mm^2^ each, along the central line were analyzed [27,28,29]. In total, 72 areas spanning the entire retina were analyzed. The average RGC density was calculated and defined as the ratio of the total number of RGCs to the entire retinal surface area.

#### 4.2.7. HE Staining

After retinal ischemia plus 1 day of reperfusion, the eyeballs were marked at the 12 o’clock position of the cornea with silk suture to ensure anatomical orientation, then enucleated and fixed in 4% paraformaldehyde at 4 °C for 24 h. After being fixed, the anterior segment was dissected, and the posterior globe along with the optic nerve head was dehydrated through a graded ethanol series and embedded in paraffin. For HE staining [30,31], 5 μm-thick sections were taken along the vertical meridian and evaluated under a light microscope (Leica, Heidelberg, Germany).

To quantitatively analyze the retinal ischemic injury, various layer thicknesses were assessed. The whole retinal thicknesses (from the inner limiting membrane to the RPE layer), and the inner retinal thickness (from the inner limiting membrane to the inner nuclear layer, INL) were evaluated. All counts were measured approximately 1 mm from the optic nerve head. Three sections per eye were averaged. Furthermore, to investigate the differences in the thickness between the four groups (sham, I/R + vehicle, I/R + 0.25 mM paeoniflorin and I/R + 0.5 mM paeoniflorin), the whole and inner retinal thickness was measured by a research staff blinded to tissue identity.

#### 4.2.8. TUNEL Assay

One day after I/R, the eyes were removed for TUNEL (In situ Cell Death Detection Kit, Fluorescein; Roche; Mannheim, Germany) to observe cell apoptosis [27]. The retinal sections were fixed with 10% formaldehyde for 24 h. Then, the retinal samples were soaked in proteinase K (25 μg/mL) and incubated with H_2_O_2_/methanol for 5 min at room temperature to inactivate endogenous peroxidases. Corresponding negative (without dUTP) and positive control (DNase-I-treated) samples were also evaluated. After washing with Tris buffered saline, the retinal sections were incubated in a TdT enzyme/labeling reaction mix at 37 °C for 90 min. This reaction was initiated by the binding of digoxigenin-dUTP to the 3′-OH end of DNA by TdT, then by soaking with an anti-digoxigenin antibody conjugated with peroxidase. Upon termination of the labeling reaction in stop buffer, the samples were processed in a standard streptavidin–horseradish peroxidase (HRP) reaction with 3,3′-diaminobenzidine as the chromogenic peroxidase substrate and were also counterstained with methyl green. The retinal samples were evaluated at 40× magnification (Zeiss, Oberkochen, Germany). Six microscopic fields from each eye, composed of three adjacent areas on both sides of the optic nerve head (ONH; 1 mm away from ONH) were examined to calculate the number of the TUNEL-positive cells in the RGC layer. The mean number of TUNEL-positive cells per field was measured for the analysis [27].

#### 4.2.9. Assessment of the Retinal VEGF/β-Catenin/HIF-1α/Ang-2 mRNA Level by Real-Time Polymerase Chain Reaction

The mRNA levels of VEGF/β-catenin/HIF-1α/Ang-2 present in the rat’s retinal cells were quantified using RT-PCR [34]. Specifically, in the normal group and the ischemic group at twenty-four hours after retinal ischemia with pre-ischemic treatment with vehicle, 0.5 mM paeoniflorin, 2 µg of PEDF, DKK, or Avastin, the rats were euthanized and the retinas were removed. This was followed by sonication in TriReagent (Sigma Chemical, St. Louis, MO, USA). Total retinal RNA was isolated and first-strand complementary DNA (cDNA) synthesis was performed on 2 μg of deoxyribonuclease (DNase)-treated RNA using the High-Capacity RNA-to-cDNA Master Mix. The first-strand cDNA then underwent real-time PCR using Fast SYBR Green Master Mix. The PCR was initiated by incubation at 95 °C for 20 s, followed by 40 cycles of 95 °C for 3 s and 60 °C for 30 s. Cycling was carried out on a StepOnePlus™ Real-Time PCR System. Relative quantification (a comparative method) was performed using the housekeeping gene β-actin or α-tubulin as the internal standard. This process allowed for the normalized quantification of the mRNA target (cycle threshold, Ct = ΔCt induced − ΔCt normal) and took into account the differences in the amount of total RNA added to each reaction (ΔCt = Ct target − Ct β-actin). The relative biomarker expression changes were calculated as fold changes relative to the control with respect to the calibrator (ΔΔCt), which was represented by the control retina. Relative quantification of gene expression was calculated according to the method 2^−ΔΔCt^, as described in the manufacturer’s instructions, and was carried out by the accompanying software (RQ, ver. 1.3). The PCR reagents, software, and machine were purchased from AB Applied Biosystems (Waltham, MA, USA). The data obtained for each treatment were pooled, and a total percentage change relative to the control was calculated. The PCR oligonucleotide primers were obtained from MISSION BIOTECH (Taipei, Taiwan) as follows: β-actin: forward primer, 5′-GAACCGCTCATTGCCGATAGTG-3′; reverse primer, 5′-TTGTCCCTGTATGCCTCTGGTCG-3′; β-tubulin: forward primer, 5′-ACATTTCAATTCGTGCTCAG-3′; reverse primer, 5′-CGTGACACCAGACATTGTGACAG-3′; HIF-1α: forward primer, 5′-AGAGCTCCCCAGCATTTCAC-3′; reverse primer, 5′-GGACAAACTCCCTCACCAAAAA-3′; VEGF: forward primer, 5′-CCTTGCCTTGCTGCTCTAC; rreverse primer, 5′-TTCTGCCCTCCTCCTTCTG-3′; Ang-2: forward primer, 5′-TCCTCCTGCCAGAGATGGAC-3′; reverse primer, 5-TGCACAGCATTGGACACGTA-3′.

### 4.3. Statistical Analysis

The graphs in Figure 1, Figure 2, Figure 3 and Figure 4 were plotted using SigmaPlot 12.5 (Jandel Scientific, Corte Madera, CA, USA), and the program SPSS 20.00 was utilized for the present statistical analysis. Statistical comparisons between groups were performed using ANOVA. If the normality test indicated a normal distribution, one-way ANOVA was employed. If the normality test failed (Shapiro–Wilk), non-parametric ANOVA was utilized. The outcomes were presented as medians and quartiles (25% and 75%). Statistical significance was established at a probability level of below 0.05 (^†^
*p* < 0.05).

## 5. Conclusions

The current in vitro OGD and H_2_O_2_ studies along with in vivo HIOP-induced I/R injury model have demonstrated that pre-treatment with 0.5 mM paeoniflorin provides significant therapeutic benefits for mitigating retinal free radicals and ischemic injury associated with RPE and RGC mortality. Importantly, further in vivo analyses suggest that paeoniflorin may protect against retinal ischemia by downregulating ischemia-related upstream β-catenin and downstream HIF-1α, VEGF, and Ang-2, with similar effects to Wnt/β-catenin inhibitor DKK1, anti-angiogenic PEDF, and anti-VEGF Avastin. Moreover, paeoniflorin reduced the I/R injury-related decline in ERG b-wave amplitudes, prevented the thinning of whole/inner retina thickness, and preserved RGC density while decreasing apoptotic cell formation. Overall, the evidence indicates that paeoniflorin confers protection against retinal ischemia and free radical injury through its neuroprotective, antioxidant, anti-angiogenic/neovascular, and potentially anti-inflammatory properties.

## Figures and Tables

**Figure 1 ijms-26-10924-f001:**
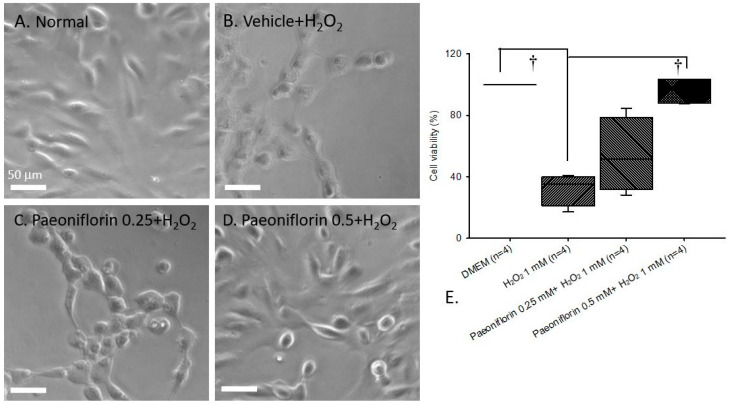
Optical microscopy was used to determine cell viability as a marker of cell death proportion. In the DMEM group (**A**), ARPE-19 cells showed the highest percentage of cell viability. In this case, free radical injury induced by the addition of 1 mM H_2_O_2_ (**B**) to ARPE-19 cells for 1 day led to a significant decline in ARPE-19 cell numbers, which was represented as a lower percentage of cell viability (**E**). In contrast to the H_2_O_2_ group (**B**), 15 min of pre-administration with 0.5 mM (**D**) paeoniflorin caused an effective protection of ARPE-19 cells (i.e., increased cell viability) against free radical injury, but the same effect was not observed with 0.25 mM (**C**) paeoniflorin. For the comparison of vehicle + H_2_O_2_ (**B**,**E**) and normal (**A**,**E**) group, 1 day incubation with H_2_O_2_ significantly (^†^
*p* < 0.05) lowered cell viability. On the other hand, 15 min of pre-treatment with 0.5 mM paeoniflorin (**D**,**E**) significantly (^†^
*p* < 0.05) neutralized the H_2_O_2_-associated (**B**,**E**) decline in cell viability, but no significant protective effects were seen at 0.25 mM of paeoniflorin. Results are represented as the median value and 25% or 75% quartiles (*n* = 4). Scale bar = 50 µm. Abbreviations: DMEM—Dulbecco’s Modified Eagle Medium; H_2_O_2_—hydrogen peroxide.

**Figure 2 ijms-26-10924-f002:**
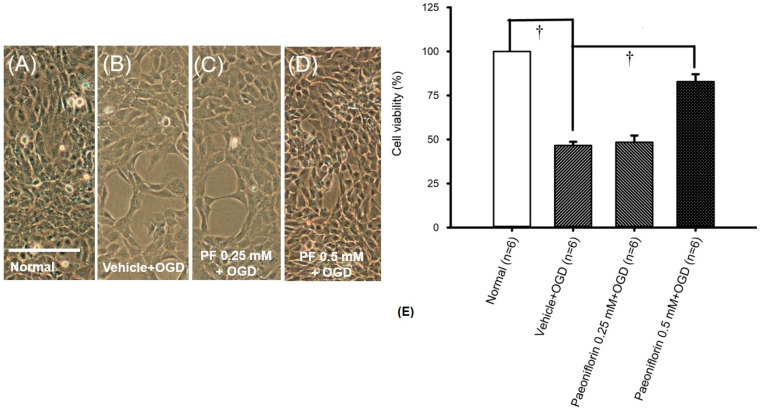
Cell viability (%) was evaluated via light microscopy as a reflection of cell survival. In the normal group (**A**), RGCs showed the highest cell count. Furthermore, oxidative stress induced by 8 h of OGD (**B**) pre-treated with vehicle (vehicle + OGD) drastically lowered cell viability (**E**). In contrast to the vehicle + OGD group (**B**), pre-treatment of RGCs with 0.5 mM (**D**) paeoniflorin for 15 min effectively protected against oxidative stress, whereas no obvious therapeutic effect was observed at 0.25 mM (**C**). In (**E**), the vehicle + OGD (**B**) group exhibited a significant (^†^
*p* < 0.05) reduction in cell viability after 8 h of OGD, compared to the normal/DMEM (**A**) group. Conversely, pre-treating for 15 min with 0.5 mM of paeoniflorin (**D**,**E**) significantly (^†^
*p* < 0.05) prevented the OGD (**B**,**E**) reduction in cell viability, while 0.25 mM of paeoniflorin offered minimal and non-significant protection. Results are represented as the mean ± SD (*n* = 6). Scale bar = 25 μm. Abbreviations: OGD—oxygen glucose deprivation.

**Figure 3 ijms-26-10924-f003:**
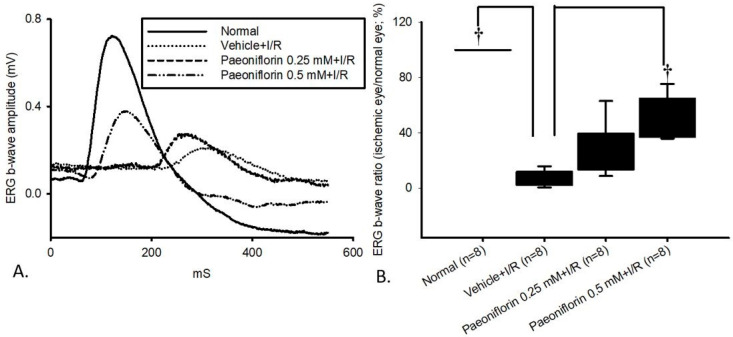
Electroretinogram (ERG): The effect of pre-ischemic intravitreous injection (IVI) of paeoniflorin on retinal I/R injury. As shown in the ERG b-wave amplitude graph (**A**), compared with the normal control retina, a significant reduction in the b-wave amplitudes was observed in the vehicle + I/R Group in Wistar rats. Attenuation of this b-wave amplitude reduction was evident after pre-ischemic administration of 0.5 mM paeoniflorin (0.5 mM paeoniflorin + I/R), but not with 0.25 mM paeoniflorin. In the ERG b-wave ratios figure (**B**), the vehicle + I/R showed a significant reduction (^†^
*p* < 0.05) at 1 day following I/R, relative to the normal group. Pre-ischemic administration of 0.5 mM paeoniflorin (0.5 mM paeoniflorin + I/R) significantly (^†^
*p* < 0.05) alleviated the ischemia-associated decline in the b-wave ratio, whereas no significant protective effect was observed in the 0.25 mM paeoniflorin group (0.25 mM paeoniflorin + I/R). Results are presented as the median value with 25% and 75% quartiles (*n* = 8). Abbreviations: I/R—ischemic/reperfusion injury.

**Figure 4 ijms-26-10924-f004:**
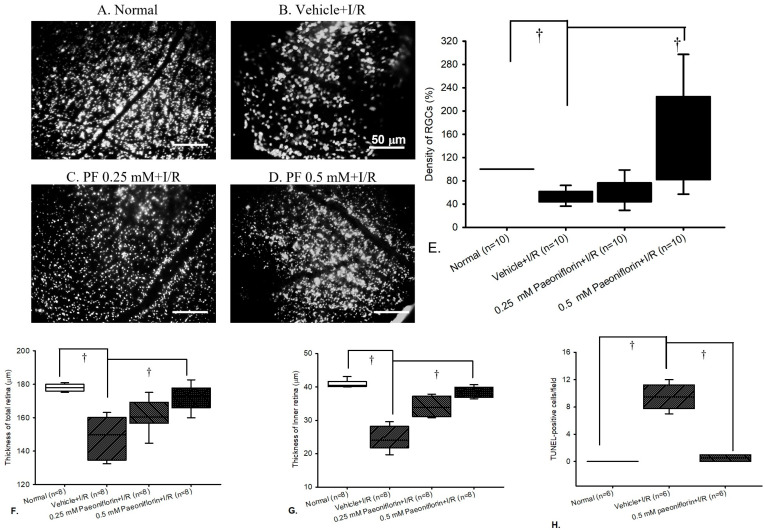
Fluorogold labeling. Microscopic images show the density of retinal ganglion cells (RGCs) in normal rats (**A**), the ischemic retina pre-treated with vehicle (**B**), and ischemic retina pre-treated with either 0.25 mM or 0.5 mM paeoniflorin (**C**,**D**). Scale bars (white) = 50 μm. As shown in the plot analysis (**E**), a statistically significant difference (^†^
*p* < 0.05) was observed between the vehicle + I/R and normal groups, indicating increased RGC mortality following I/R injury. On the other hand, pre-treatment with 0.5 mM paeoniflorin provided a protective effect, as evidenced by a significant difference (^†^
*p* < 0.05) between the vehicle + I/R and 0.5 mM paeoniflorin + I/R groups. However, pre-treatment with 0.25 mM paeoniflorin did not produce a significant therapeutic effect on RGC loss induced by I/R. Each bar represents the median value and 25% and 75% quartiles (*n* = 10). Retinal layer thickness. Morphometric analysis of the whole and inner retinal layers was performed using sections of similar eccentricity (**F**,**G**). Compared with Group 1 (normal) both whole retinal and inner retinal layers were significantly reduced in the vehicle-treated ischemic retinas (Group 2). These reductions were alleviated in a dose-dependent manner by pre-treatment with 0.25 mM (Group 3) and 0.5 mM paeoniflorin (Group 4). Results are expressed as the mean ± SD (*n* = 8). Significant differences represented as (^†^
*p* < 0.05). TUNEL assay. Quantification of apoptotic cells (TUNEL-positive cells) across different experimental groups (**H**) revealed a significant increase (^†^
*p* < 0.05) in the number of apoptotic cells for the vehicle + I/R group. Pretreatment with 0.5 mM paeoniflorin significantly (^†^
*p* < 0.05) reduced apoptotic cell counts, relative to the vehicle + I/R. Results are median, 25% and 75% quartiles (*n* = 6). Abbreviations: I/R—ischemic/reperfusion injury.

**Figure 5 ijms-26-10924-f005:**
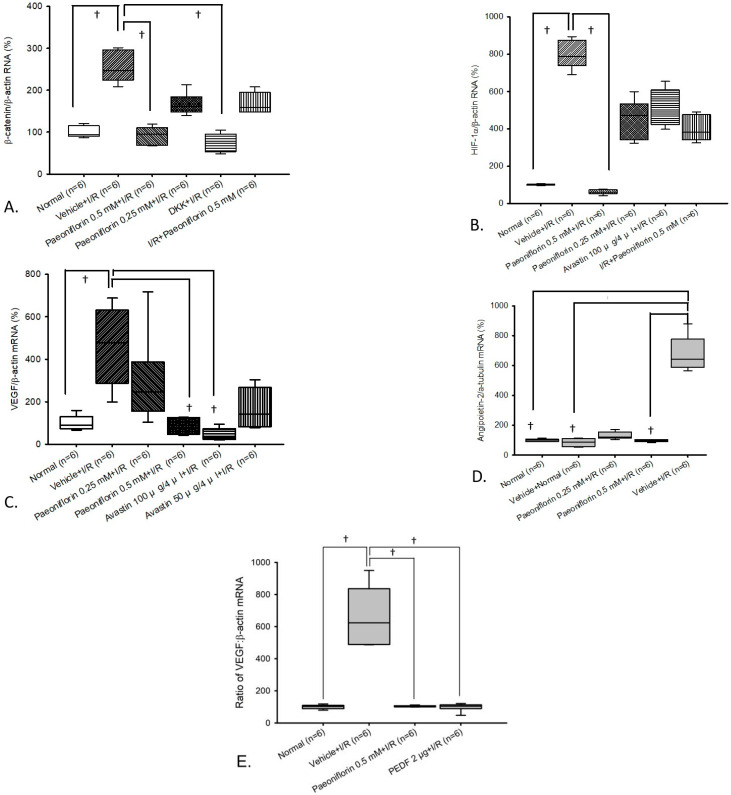
(**A**–**D**) β-catenin, HIF-1α, VEGF, or Ang-2 mRNA expression ratios were analyzed using real-time polymerase chain reaction (RT-PCR). In each figure, target mRNAs were normalized to the housekeeping biomarker β-actin or α-tubulin. Column 1 shows the normal group; column 2 displays the pre-ischemic vehicle with I/R group (except for (**D**), where column 5 is shown); columns 3 and 4 show the I/R retinas 15 min pre-administered with 0.25 or 0.5 mM of paeoniflorin; the remaining columns contain ischemic retinas pre-administered with DKK (**A**) versus 50 or 100 μg/4 μL of Avastin (**B**,**C**), corresponding to the mRNAs studied. In the vehicle + I/R group, the retinas showed a notable (^†^
*p* < 0.05) increase in β-catenin, HIF-1α, VEGF, and Ang-2 mRNA concentrations, relative to the normal (control) group. The elevation of these ischemic/angiogenic/inflammatory-related biomarkers was significantly (^†^
*p* < 0.05) reduced by 15 min of pre-treatment with 0.5 mM paeoniflorin (**A**–**D**), DKK (**A**), and 100 μg/4 μL of Avastin (**C**). Notably, pre-treatment with 0.25 mM paeoniflorin (**A**–**D**) and 50 μg/4 μL of Avastin (**B**,**C**) did not produce significant downregulation of these mRNAs. Results are represented as the median value and 25% and 75% quartiles (*n* = 6). (**E**) RT-PCR analysis of VEGF mRNA levels. Retinas were evaluated 24 h after HIOP-induced I/R, revealing a significant (^†^
*p* < 0.05) increase in VEGF levels compared to the normal (control) group. This contrasts with the low baseline VEGF levels observed in the normal group. The significant elevation was attenuated (^†^
*p* < 0.05) by 15 min pre-treatment with 0.5 mM paeoniflorin, showing effects similar to those of anti-angiogenic factor PEDF. Each bar is represented as the median value and 25% and 75% quartiles (*n* = 6). Abbreviations: I/R—ischemic/reperfusion injury; DKK—dickkopf-related biomarker; PEDF—pigment epithelium-derived factor.

## Data Availability

The original contributions presented in this study are included in the article. Further inquiries can be directed to the corresponding author.

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
