# Peer review of "Retinal Ischemia: Therapeutic Effects and Mechanisms of Paeoniflorin"

_ijms, 2025, doi:10.3390/ijms262210924_

Round 1
Reviewer 1 Report
Comments and Suggestions for Authors
Major revision
The current manuscript lacks mechanistic insights into how paeoniflorin downregulates β-catenin, HIF-1α, VEGF, and the pro-inflammatory/angiogenic marker angiopoietin-2. It is essential to identify the upstream molecular targets or signaling pathways involved in this regulatory effect to establish the therapeutic relevance of paeoniflorin.
In addition, the in vitro experiments exclusively utilized RPE cells, which are epithelial in nature. However, the retina is composed of a complex multicellular network including endothelial cells, pericytes, astrocytes, microglia, and neurons. The use of RPE cells alone does not sufficiently represent overall retinal cell viability. Therefore, the conclusion that paeoniflorin improves “retinal cell viability” based solely on RPE cell data is not adequately supported.
Furthermore, the connection between the observed increase in RPE cell viability in vitro and the multiple in vivo outcomes—such as preservation of ERG b-wave amplitude, retinal thickness, and RGC survival—is not clearly explained. It is difficult to understand how enhanced RPE survival directly translates to these diverse retinal structural and functional improvements. Additional data or rationale is needed to clarify the relevance of RPE protection in the context of the broader retinal ischemic injury model
Minor revision
- (Page 3, line 3):
There is a typographical error in the term “free radicle.” It should be corrected to “free radical” to accurately reflect the intended meaning related to oxidative stress. - (Page 3, line 16):
The sentence describing the histopathological methods (“Additional histopathological methods included retrograde fluorogold RGC immunolabelling, hematoxylin and eosin (HE) staining and terminal deoxynucleotidyl-transferase (TdT)-mediated dUTP nick end-labeling (TUNEL) assay.”) appears to be repeated. Please remove the redundancy to improve clarity and conciseness. - (Page 8):
In the phrase “which labelled as the Vehicle+I/R group,” the verb is missing an auxiliary and needs correction. It should be revised to “which was labeled as the Vehicle+I/R group” to ensure proper grammatical structure. - (Page 8):
The phrase “a significant (†P < 0.05) effect was seen at higher the concentration of 0.5 mM paeoniflorin” contains a grammatical error. Please revise it to “at the higher concentration of 0.5 mM paeoniflorin” or more naturally to “at 0.5 mM paeoniflorin” for improved clarity and correctness. - There is a need for consistent formatting of statistical significance throughout the manuscript. For example, the expressions “significantly (*P < 0.05)” and “significantly (†P < 0.05)” are both used; it is recommended to unify these notations to maintain clarity and professionalism.
- Throughout the manuscript, the use of abbreviations and full terms is inconsistent and should be carefully reviewed. All abbreviations (e.g., I/R, VEGF, PEDF) must be defined upon first mention and used consistently thereafter, in accordance with standard scientific writing conventions. Please revise the manuscript to ensure that all abbreviations are appropriately introduced and formatted.
- There are multiple grammatical and stylistic errors throughout the manuscript that hinder clarity and readability. It is strongly recommended that the manuscript be professionally proofread by a native English speaker or a scientific editing service prior to resubmission.
Author Response
Please see the attached word document below.

Reviewer 2 Report
Comments and Suggestions for Authors
The manuscript "Retinal ischemia: therapeutic effects and mechanisms of paeoniflorin" by Chao and coworkers describes the evaluation of paeoniflorin for its capability to modulate biological phenomena associated with retinal ischemia. The authors employed both cell culture and animal models, which is of great interest to expand the knowledge about natural products. The manuscript contains sufficient theoretical background and discussion to compare the retrieved data with other studies, where the effect of paeoniflorin has also been evaluated for retinal ischemia treatment, considering other molecular pathways. Before accepting it for publication, the authors are recommended to address the following comments.
-The authors are recommended to add, if possible, the epidemiology of retinal ischemica cases in the introduction. This with the purpose of highlighting the need of addressing treatment approaches towards populations affected by this condition.
-The introduction is missing to explain the importance of natural products and why they must be considered for treatment retinal ischemia. Considering this, the authors are recommended to mention the chemical structure of paeoniflorin, why is it studied, what are its sources, and how it can be extracted. If the biological performance varies among derivatives, the authors are recommended to include similar aspects.
-The materials and methods section is missing to explain what was the source of paeoniflorin either if it was isolated, extracted, or purchased. In the case of isolation or extraction, the authors are advised to explain this in detail.
-The panels of Figure 1 must be revised since the text is not in the same font, and it seems that it is written with different types of letters. If it is a matter of size, the text can be moved to the upper part of the image. In panel B of the same figure, the authors must revise the use of lower scripts in "H2O2" on the y-axis. The same comment must be addressed in the legend of Figure 1 and the conclusion section.
-Revise the use of italics in words such as "in vitro" and "in vivo" throughout the manuscript.
-Various references are outdated. The authors are recommended to update them using research articles from high impact journals.
Author Response

(The authors gave the same response as above.)

Round 2
Reviewer 1 Report
Comments and Suggestions for Authors
It has improved overall.
Comments on the Quality of English LanguageIt has improved overall.
Reviewer 2 Report
Comments and Suggestions for Authors
The authors have addressed all the comments. It is recommended for publication.